# Five Rice Seed-Specific *NF-YC* Genes Redundantly Regulate Grain Quality and Seed Germination via Interfering Gibberellin Pathway

**DOI:** 10.3390/ijms23158382

**Published:** 2022-07-29

**Authors:** Huayu Xu, Shufan Li, Bello Babatunde Kazeem, Abolore Adijat Ajadi, Jinjin Luo, Man Yin, Xinyong Liu, Lijuan Chen, Jiezheng Ying, Xiaohong Tong, Yifeng Wang, Baixiao Niu, Chen Chen, Xiaoshan Zeng, Jian Zhang

**Affiliations:** 1State Key Lab of Rice Biology, China National Rice Research Institute, Hangzhou 311400, China; xuhuayu202107@163.com (H.X.); shufanli1994@163.com (S.L.); tunlapa2k13@gmail.com (B.B.K.); threetriplea@yahoo.com (A.A.A.); luojinjin998220@163.com (J.L.); 13545663721@163.com (M.Y.); liuxinyong1234@gmail.com (X.L.); chenlijuan0723@163.com (L.C.); yingjiezheng@caas.cn (J.Y.); tongxiaohong@caas.cn (X.T.); wangyifeng@caas.cn (Y.W.); 2College of Agriculture, Yangzhou University, Yangzhou 225009, China; bxniu@yzu.edu.cn (B.N.); chenchen@yzu.edu.cn (C.C.); 3Hunan Rice Research Institute, Hunan Academy of Agricultural Sciences, Changsha 410125, China

**Keywords:** rice (*Oryza sativa* L.), gibberellins, abscisic acid, NF-YCs

## Abstract

NF-YCs are important transcription factors with diverse functions in the plant kingdoms including seed development. *NF-YC8*, *9*, *10*, *11* and *12* are close homologs with similar seed-specific expression patterns. Despite the fact that some of the *NF-YCs* are functionally known; their biological roles have not been systematically explored yet, given the potential functional redundancy. In this study, we generated pentuple mutant *pnfyc* of *NF-YC8-12* and revealed their functions in the regulation of grain quality and seed germination. *pnfyc* grains displayed significantly more chalkiness with abnormal starch granule packaging. *pnfyc* seed germination and post-germination growth are much slower than the wild-type NIP, largely owing to the GA-deficiency as exogenous GA was able to fully recover the germination phenotype. The RNA-seq experiment identified a total of 469 differentially expressed genes, and several GA-, ABA- and grain quality control-related genes might be transcriptionally regulated by the five NF-YCs, as revealed by qRT-PCR analysis. The results demonstrated the redundant functions of *NF-YC8-12* in regulating GA pathways that underpin rice grain quality and seed germination, and shed a novel light on the functions of the seed-specific *NF-YCs*.

## 1. Introduction

Rice (*Oryza sativa* L.) is a major cereal crop in the world, as it is consumed as a staple food by more than half of the world’s population [1]. Rice seed is a complex organ that is comprised of a maternal caryopsis coat, a diploid embryo and a triploid endosperm. The nutrients such as starch, protein and lipids are accumulated in the endosperm underpinning seed germination or grain yield and quality for human consumption. It has been known that phytohormones are extensively involved in the regulation of plant seed development [2,3,4,5]. The action of GAs and ABA on seed development is strictly correlated and antagonistic [2]. Through the investigation of the rice seed hormonal dynamics during the grain filling stage, Yang et al. (2001) revealed that GAs play key roles in embryogenesis, while the ABA content reached the peak at a much later stage, thus it seemed to be more relevant to the seed maturation [6]. So far, numerous pieces of literature about genes controlling rice seed development have been published, and these genes are involved in transcriptional regulation, the ubiquitin–proteasome pathway, plant hormone response, and so on [7,8,9]. Specifically, Yao et al. (2017) found that *NF-YC8* to *NF-YC12* are five important genes involved in starch synthesis, seed storage protein and the stress response [7]. The study shows that *NF-YC12* is a key transcription factor in regulating endosperm development [8] and storage material accumulation in rice seeds [10]. A novel transcription factor subunit *NF-YC13* was identified in indica rice, which can respond to salt stress signals by interacting with the B-subunit [11]. It is studied that NF-YC2 and NF-YC4 proteins can interact with three flowering-time genes to regulate the photoperiodic flowering response under long sunlight conditions [12].

Nuclear Factor Y (NF-Y) is a family of transcription factors that are found in vast quantities in higher eukaryotes. The NF-Y protein complex consists of three subunits: NF-YA (CBF-B/HAP2), NF-YB (CBF-A/HAP3), and NF-YC (CBF-C/HAP5) which usually forms a heterotrimer to regulate the transcription of the target genes [13,14]. For yeast and animals, each NF-Y subunit is encoded by a single gene. However, the situation in plants is more complicated with multiple members of each subunit, which dramatically expanded the diversity of *NF-Y*s’ gene function in the plant kingdom [13,15]. In rice, each NF-Y subunit covers more than 10 gene members as reported, and many of them have been identified to participate in extensive developmental processes like nutrient accumulation in the endosperm, flowering regulation and ABA signal response [16,17,18,19]. *NF-YBs* are well-documented among those three subunits and have been implicated in plant height regulation, grain yield, carbon assimilation, photoperiodic flowering and other processes [20,21,22,23,24]. For example, *NF-YB2*, *NF-YB3* and *NF-YB4* are close homologs that are functionally redundant in regulating chloroplast biogenesis in rice [25]. It is noteworthy that several *NF-YBs* and *NF-YCs* were found to be specifically expressed in rice endosperm. Some seed-specific *NF-YBs* control rice seed development by affecting the nutrition accumulation and the loading of sucrose into developing seeds [26,27,28]. In addition, endosperm-specific *NF-YBs* and *NF-YCs* may also form heterotrimer complexes with other non-NF-Y transcription factors, hence regulating grain filling and quality via the ubiquitin–proteasome pathway [8,29]. For example, AtNF-YB9-YC12-bZIP67 can activate the expression of *SUS2* and promote seed development [30]; OsNF-YB1 interacts with OsNF-YC12, and OsNF-YC12 can bind to the promoter of *FLO6* and *OsGS1;3* to regulate grain weight and chalky endosperm [10].

Several previous works have revealed that OsNF-YC8 (LOC_Os01g01290), OsNF-YC9 (LOC_Os01g24460), OsNF-YC10 (LOC_Os01g39850), OsNF-YC11 (LOC_Os10g23910) and OsNF-YC12 (LOC_Os05g11580) are close homologs with similar seed-specific expression pattern, implying that they play a role in rice seed development [7,8,29,31]. So far, it is known that the NF-YC9 controls cell proliferation to influence grain width, and NF-YC11 regulates the accumulation of storage substances in rice seeds [10,32]. In 2019, our lab reported that NF-YC12 forms a heterotrimer complex with NF-YB1 and bHLH144 to regulate rice grain quality [8]. Nevertheless, knowledge about the function of the five genes is rather fragmented, given that the high similarity of the genes may give rise to functional redundancy. Here, we report the systematic functional analysis of NF-YC8, 9, 10, 11 and 12 using single gene or pentuple gene mutants. The five genes may work redundantly to regulate ABA and GA response, thus determining grain quality and seed germination. This work sheds new insight into the functional roles of the seed-specific OsNF-YCs. 

## 2. Results and Discussion

### 2.1. Five Seed-Specific NY-YCs Works Redundantly to Regulate Grain Quality

To specify the biological functions of *OsNF-YC8*, *OsNF-YC9*, *OsNF-YC10*, *OsNF-YC11* and *OsNF-YC12*, we generated a single mutant of each gene in the background of Kitaake (*Oryza sativa* ssp. Japonica), respectively. Sanger sequencing further confirmed the mutations of the corresponding genes with insertion or deletions, which should have shifted the open reading frame and disrupted the resulting protein functions (Appendix A). Compared with Kitaake, *nfyc8* exhibited increased percentage of grains with chalkiness (PGWC), *nfyc12* had higher degree of chalkiness (DEC), while *nfyc9* and *nfyc10* exhibited increased PGWC and higher DEC (Appendix A). We used the CRISPR/Cas9 technology to simultaneously knock out all the five *NF-YC* genes in Nipponbare (NIP, *Oryza sativa* ssp. japonica) to generate the pentuple *nf-yc* mutants (hereafter referred to as *pnfyc*) to assess the potential functional redundancy among the genes. Two representative lines *pnfyc-1* and *pnfyc-2* were selected for the followed genotyping and genetic analysis. As shown in Appendix A, Sanger sequencing detected various types of homozygous insertion or deletion mutations in each of the *NF-YC8-12*, suggesting all the five genes were successfully knocked-out. During the vegetative growth stage, no visible differences were observed in major agronomic traits such as plant height, flowering date, seed setting and spikelets per panicle in the *pnfyc* lines (Appendix A). However, the milled grains of *pnfyc* lines showed obvious chalkiness. As revealed by the cross-sections of the *pnfyc* seeds, the starchy endosperm of *pnfyc* was floury-white when compared with NIP. Scanning electron microscopy (SEM) images of transverse sections indicated that the starch granule of NIP and *pnfyc* grains had different morphologies, shape and packaging densities. Unlike the regular shape of the starch granule of NIP, *pnfyc* had irregular, loosely packed starch granules, which might be responsible for the observed chalkiness (Figure 1A). Furthermore, we examined the contents of storage substances in the brown seeds, and found that the total starch and amylose contents of *pnfyc* were significantly lower than that of the NIP. Conversely, *pnfyc* had relatively higher crude protein contents than NIP (Figure 1B–D). The PGWC in *pnfyc* reached over 90%, while that of NIP was less than 10% (Figure 1E). Following the change in the starch contents, differential scanning calorimetry (DSC) analysis demonstrated that the gelatinization characteristics including the onset, peak as well as end gelatinization temperatures of *pnfyc* were also significantly altered (Figure 1F and Appendix A). The results above indicated that *OsNF-YC8*, *9*, *10*, *11* and *12* work redundantly to positive regulate rice grain quality, particularly the grain chalkiness.

### 2.2. pnfyc Are GA-Deficient with Retarded Seed Germination

The altered storage substance proportion in *pnfyc* lines provoked us to test the roles of the five *NF-YCs* in seed germination. We carried out germination assays on the *nfyc* single mutants, *pnfyc* and NIP lines on ½ MS medium for 4 days. All the five *nfyc* single mutants showed slightly lower germination rates compared with Kitaake, which was followed by retarded post-germination growth (Appendix A). However, the *pnfyc* seeds showed much retarded germination (56.7–60.0%) than the NIP (90.0–96.7%) at day 4, which further confirmed the functional redundancy among the five NF-YCs. Given the key roles of ABA and GA in seed germination regulation, we subsequently investigated the seed germination under exogenous ABA, GA and PAC (GA biosynthesis inhibitor) treatments (Figure 2A–C). The application of 2 μM exogenous ABA significantly restrained the seed germination of both *pnfyc* and NIP seeds. To evaluate the relative ABA sensitivity of the seeds, we calculated the relative germination rate of the seeds under mock and ABA treatments. The results showed that the WT relative germination rates of 2 μM ABA/mock and 5 μM ABA/mock were 72.2% and 65.5%, respectively. However, for the *pnfyc* seeds, 84.5% and 51.4% were obtained. Therefore, *pnfyc* is hypersensitive to exogenous ABA treatment in seed germination (*p* < 0.05) (Figure 2D). Similar hypersensitivity to ABA inhibition effects was also observed in the post-germination growth of *pnfyc* seedlings. In contrast to the ABA treatments, the application of 2 μM exogenous GA significantly recovered the seed germination of *pnfyc*, and a more intense recovering effect was observed when 5 μM exogenous GA was applied, suggesting the GA deficiency might be the major reason for the retarded seed germination in *pnfyc*. Exogenous PAC displayed similar inhibitory effects as ABA, as 10 μM exogenous PAC decreased the NIP germination rate and post-germination growth to the *pnfyc* level. In addition, we quantified the GA3 contents in the germinative seeds of NIP and *pnfyc-1*. The result showed that the GA3 content in *pnfyc-1* was reduced to only 30% of the NIP (*p* < 0.01), which is in agreement with the recovered germination of *pnfyc* by GA (Figure 2E).

To test the involvement of the five *NF-YCs* in ABA and GA biosynthesis and signaling, we examined the transcription of the genes in the germinative seeds of various ABA or GA-related genetic lines by qRT-PCR. *SAPK8*, *9* and *10* are core elements of ABA signaling, and over-expression of the genes conferred plants ABA hypersensitivity [33]. We found that *NF-YC10* was significantly up-regulated by *SAPK8* and *10*, while *NF-YC8* and *NF-YC12* were up-regulated by *SAPK10* and *SAPK9*, respectively. However, *NF-YC9* and *NF-YC11* were down-regulated in all the three *OxSAPK* lines (Figure 3A). In the GA-deficient mutant *sd1* [34], the transcription of *NF-YC8*, *9*, *10* and *11* were all severely repressed, indicating the five *NF-YCs* are highly responsive to endogenous GA level (Figure 3B). Taken together, we proposed that *NF-YC8*, *9*, *10*, *11* and *12* may serve as key regulators mediating the balance of GA and ABA.

Although *NF-YC10* and *NF-YC12* have been reported as key regulators of rice seed development [8,10,32], the roles of the five seed-specific NY-YCs are still not very clear so far, given the potential functional redundancy among them. By simultaneously knocking out the five genes, we revealed their functions in grain quality and seed germination, and the GA-deficiency in *pnfyc* might be the major reason for the observed phenotype. Aside from the well-known function as a seed germination promoter, GA has been recently found to regulate endosperm development as well. Cui et al. (2020) reported that application of exogenous GA_4+7_ significantly altered the content of other phytohormones such as auxin, zeatin and ABA, increased the activities of superoxide dismutase, catalases, and peroxidases and reduced the malondialdehyde content, which finally improved grain filling and yield in maize [35]. In Arabidopsis, *NF-YC3*, *NF-YC4* and *NF-YC9* have redundant roles in the regulation of GA-ABA-mediated seed germination [36]. NF-YCs bind RGL2, a repressor during GA signaling, and then in the form of NF-YC–RGL2 module targets *ABI5*, a key factor in ABA signaling [37,38]. NF-YC can bind to the CCAAT-box on the *ABI5* promoter to regulate ABI5 gene expression. The NF-YC–RGL2–ABI5 module integrates ABA and GA signaling to regulate seed germination [36].

### 2.3. NF-YCs Regulates the Transcription of ABA and GA Pathway Genes

RNA sequencing experiments on the 6 HAI (6 h after imbibition) germinating seeds of *pnfyc* and WT were carried out to identify the potential target genes of the five NF-YCs. As a result, there were a total of 469 differentially expressed genes (DEGs) as shown in (Appendix A). KEGG analysis revealed predominant enrichment of the DEGs on the pathways of ‘phenylpropanoid biosynthesis’, ‘protein processing in endoplasmic reticulum’ and ‘plant hormone signal transduction’ (Appendix A). We further carried out a qRT-PCR analysis on eight randomly selected DEGs to verify the RNA-seq results, and the results showed that seven of those genes had a similar transcriptional level inclination to that detected by the RNA-seq, indicating the high reliability of our RNA-seq results (Appendix A). Notably, a series of genes reported as critical regulators of the ABA signal pathway were found to be down-regulated in *pnfyc* (Figure 4B), including positive ABA signaling factors like *OsbZIP46* [39,40], *OsbZIP12* [41,42] and *OsNAC52* [43] as well as the *OsHSP24.1* encoding an ABA-responsive heat shock protein [44]. Furthermore, we conducted the qRT-PCR and results showed that the expression of most GA biosynthesis genes was down-regulated in *pnfyc* plants, while the expression of ABA biosynthesis and negative signal pathway-related genes was up-regulated (Figure 4A,B).

Given the severely affected starch qualities in *pnfyc* seeds, we also examined the transcriptional levels of several starch biosynthesis enzyme or regulator genes in the developing seeds of *pnfyc* and NIP. It was found that, except for *ISA2*, all of these ADP-glucose pyrophosphorylase, granule-bound starch synthase, starch synthase and starch branching enzyme were mostly down-regulated in 7 DAP endosperm of *pnfyc* lines (Figure 4C) [45,46,47,48].

### 2.4. Potential Interactive Proteins of the Five NF-YCs

We tested the protein–protein interaction between the 5 NF-YCs and 10 SAPKs which are ABA signaling components. A total of 50 NF-YC-SAPK combinations were tested by yeast-two-hybrid, and results showed that NF-YC10 binds to SAPK4, 6 and 10, while all the other combinations were negative. Hence, the suggestion is that NF-YC10 perceives the ABA signal from SAPK4, 6 and 10 (Figure 5A).

Our previous study has demonstrated that NF-YB1-YC12 dimer binds to bHLH144 to form a heterotrimer complex that regulates rice grain quality [8]. To identify other components that may interact with NF-YB1-YC12 dimer, we performed yeast-three hybrid (Y3H) experiments to screen a seed-derived prey library using NF-YB_1_-YC_12_-pBRIDGE as bait. We finally obtained three interactive proteins LOC_Os01g68950, LOC_Os07g46160 and LOC_Os11g38670 which are annotated as ubiquitin domain-containing protein, BTB/POZ domain-containing protein and dead-box ATP-dependent RNA helicase, respectively. Interestingly, the interactions are valid only on the SD/-Met/-Leu/-Trp/-Ade/-His/+X-α-Gal medium, in which the drop-out of methionine drove the expression of NF-YC12 under Met25 promoter. Meanwhile, the interactions were compromised on SD/-Leu/-Trp/-Ade/-His/+X-α-Gal medium, in which NF-YC12 was suppressed by the supplemented methionine in the medium. Hence, the binding of NF-YB1-YC12 is necessary for the formation of heterotrimer complexes with the three proteins (Figure 5B–D).

In conclusion, we report a rice pentuple gene mutant *pnfyc,* which knocked out five homologous genes *OsNF-YC8, OsNF-YC9, OsNF-YC10, OsNF-YC11* and *OsNF-YC12* simultaneously. The expression of starch synthesis genes decreased in *pnfyc,* resulting in the decrease of starch content and the increase of protein content, the change of grain quality and the significant increase of chalkiness trait. The results showed that *NF-YC8* to *NF-YC12* could regulate grain quality traits by regulating starch synthesis. In addition, *NF-YC8-12* also inhibited seed germination by affecting the expression of GA-related genes, and the phenotype is significantly restored by applying exogenous GA. Finally, the expression of ABA-related genes in *pnfyc* increased, and *pnfyc* seeds were hypersensitive to exogenous ABA. NFYC10 could interact with SAPK to regulate ABA expression.

## 3. Materials and Methods

### 3.1. Plant Growth Conditions and Phenotype Measurement

To generate the CRISPR/Cas9-derived knock-out mutants, the specific target small-guide RNA (sgRNA) of each gene was designed and assembled in a pYLCRISPR/CAS9-MH vector system according to a previous report [49], and subsequently transformed into Nipponbare and Kitaake (*Oryza sativa*, ssp. japonica) backgrounds. All the plants were grown in the experimental greenhouse and field of the China National Rice Research Institute (CNRRI). Agronomic traits were analyzed with 10 replicates. Panicle length, number of primary branch panicles, number of effective panicles per plant, seed setting rate (%), and plant height were measured manually. Rice seed grains were harvested and air-dried at room temperature for at least 2 weeks. The thousand-grain-weight, seed length, width and chalkiness were examined by a seed phenotyping system (Wan Sheng, Hangzhou, China). Grain thickness was determined at the same time for each grain using an electronic digital calliper.

### 3.2. Physicochemical Properties of Seed Grain

Total starch content of the dried brown seeds was measured using a starch assay kits Megazyme K-TSTA and KAMYL (Megazyme, Ireland, UK, http://www.megazyme.com/ accessed on 6 May 2020). The total amylose and protein contents in the grains were measured by following a previous report [50]. The content is expressed as the percentage of total sample weight on an oven-dry basis. To analyze the gelatinization temperature, DSC assay was conducted on a differential scanning calorimeter DSC1 STARe system (METTLER-TOLEDO, Zurich, Switzerland). Briefly, 5 mg rice powder was sealed and placed in an aluminum sample cup, mixed with 10 μL distilled water, and then the samples were analyzed by the differential scanning calorimeter (METTLER-TOLEDO, Zurich, Switzerland). The heating rate was 10 °C min^−1^ over a temperature range of 40 °C to 100 °C [51].

### 3.3. Scanning Electron Microscopy (SEM) Assay

Prepare two types of milled rice, one was wild-type NIP and the other was the *pnfyc* mutant. The whole grains were cut transversely with a sharp blade and then sputtered with gold in order to increase electrical conductivity. Fractured rice grains were mounted on the copper stage and then viewed with a scanning electron microscope at 30, 2000 and 5000 times magnification. The analysis was performed based on three biological replicates at least. The experiment was conducted in institute of Agriculture and Biotechnology, ZheJiang University as described previously using a HITACHI S-3400N scanning electron microscope (HITACHI, Tokyo, Japan).

### 3.4. Seed Germination and Phenotypic Assay

Briefly, 100 dehusked seeds were surface-sterilized in 75% ethanol for 2 min, then in 50% bleach for 30 min, and then cleaned with sterilized ddH_2_O 5–8 times for 3 min each time. The sterilized seeds were air dried and sown on a ½ strength MS medium containing different concentrations of ABA (0, 2, 5 µM), GA (0, 2, 5 µM) and PAC (0, 2, 5, 10 µM). Germination rates were recorded every 12 h. The seedlings’ height above ground was measured and the growth status of seedlings was photographed after 7 days. Germination is established with the appearance of the emergence of 2 mm embryos through the seed coat. The data are the mean of 3 biological triplicates.

### 3.5. RNA-Seq Analysis and RT-PCR Analysis

For RNA-seq, total RNA of germinative seeds at 6 h-after-imbibition was extracted using Trizol as instructed (Yeasen, Shanghai, China). The high-throughput sequencing was performed using the Illumina HiSeq™ 2500 platform and the KEGG pathway analysis of the DEGs was ultimately done by Personalgene Technology Co (Personal, Shanghai, China). DEGs were defined as genes with |log2Fold change| ≥ 1 and FDR < 0.01 using EBSeq [52]. The endosperm of 7 days after fertilization and seeds of 6 h after germination were collected as samples for the extraction of RNA to detect the expression level of genes in starch biosynthesis and hormone synthesis, and the RNA was extracted by Trizol according to the kit manufacturer’s instructions (Yeasen, Shanghai, China). 

For the RT-PCR analysis, the first-strand cDNA was synthesized using M-MLV reverse transcriptase according to the manufacturer’s instructions (Takara, Dalian, China). The expression levels of different samples were determined using CFX96 touch real-time PCR detection system (Bio-Rad, Hercules, CA, USA). Expression was assessed by evaluating threshold cycle (CT) values. The relative expression level of tested genes was normalized to ubiquitin gene and calculated by the 2^−∆∆CT^ method. The experiment was performed in two biological replicates with three technical triplicates of each. Primer sequences are listed in Appendix A.

### 3.6. Yeast-Two-Hybrid Assay

The yeast two-hybrid assay was conducted based on the manufacturer’s protocol (Invitrogen, Carlsbad, CA, USA). The coding sequence of *NF-YC8* to *NF-YC12* and *SAPK1* to *SAPK10* were amplified and cloned into bait vector pdest32 and prey vector pdest22, respectively. Two vectors, pDEST32-NFYCs and pDEST-SAPKs, were co-transfected into the Y2H Gold strain, and then yeast cells were grown on SD/-Trp/-Leu and SD/-Trp/-Leu/-His/-Ade medium for screening. pGBKT7-53 and pGADT7-T were used as positive controls, pGBKT7-Lam and pGADT7-T were used as negative controls. The primers used are listed in Appendix A.

### 3.7. Yeast-Three-Hybrid Assay

*NF-YB1* CDS was cloned to fuse with GAL4 BD domain, and NF-YC12 was driven by a methionine-responsive promoter Met25 in pBRIDGE (Clontech, Dalian, China). NF-YB1-NF-YC12-pBRIDGE in strain Y2H Gold was mated with an AD domain-fused seed cDNA library in Y187 strain. The mated transformants were first selected on SD/-Leu/-Trp. Positive colonies were then transferred to SD/-Leu/-Trp/-His/-Ade/-Met/+X-a-Gal and SD/-Leu/-Trp/-His/-Ade/+X-a-Gal, respectively. The interaction was confirmed by the visualization of blue colonies on the medium.

## Figures and Tables

**Figure 1 ijms-23-08382-f001:**
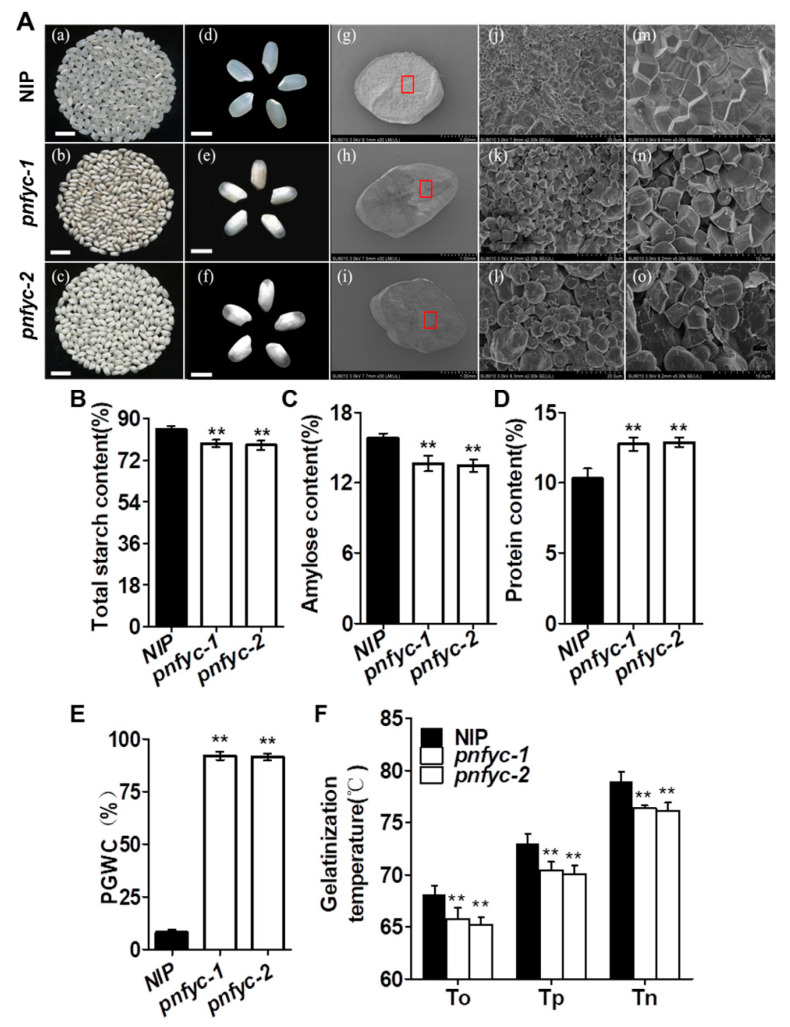
(**A**) (a–o) Grain chalkiness phenotypical characterization (a–f), bar = 2 mm. Scanning electron microscopy (SEM) analysis (g–o). The central areas shown are indicated as red squares. The magnification is 30 times in (g,h,i); 2000 times in (j,k,l), and 5000 times in (m,n,o). (**B**–**F**) Quality trait parameters of mature seeds from *pnfyc* lines and NIP. Data are shown as means ± SD of at least three biological replicates. (** *p* < 0.01 by two-tailed Student’s *t*-test).

**Figure 2 ijms-23-08382-f002:**
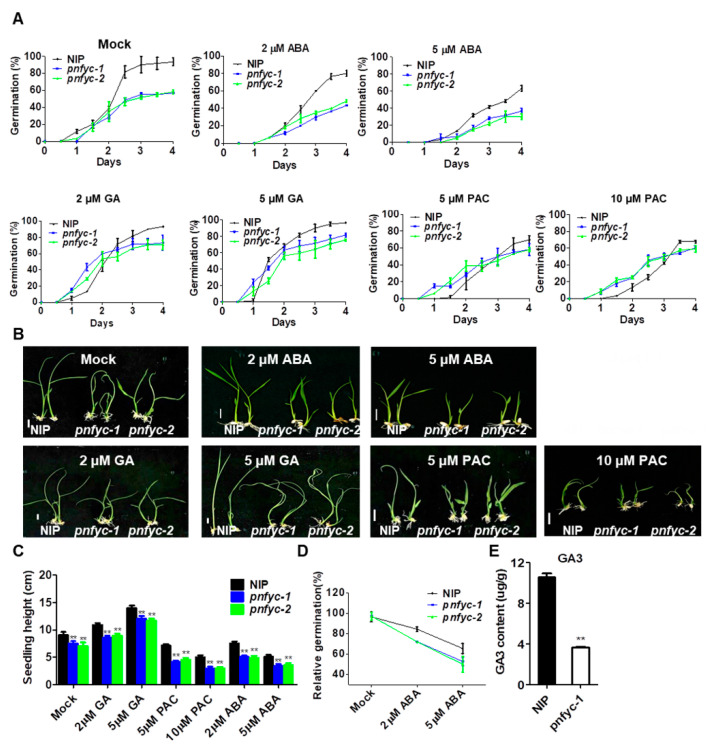
(**A**) The germination rate of NIP and *pnfyc* lines under different concentrations of exogenous hormones. (**B**) The plant morphology of NIP and *pnfyc* lines treated with different exogenous hormones at 7 days after germination. (**C**) The seedling height of NIP and *pnfyc* lines under different exogenous hormones at 7 days after germination. Bar = 1 cm. (**D**) The relative germination of the NIP and *pnfyc* seeds under ABA treatments were determined after 4 days and expressed as a percentage of those grown under ‘mock’ conditions. (**E**) Quantification of GA3 derivatives in NIP and *pnfyc* seeds germinated for 6 h was analyzed with liquid chromatography-tandem mass spectrometry. Data are shown as means ± SD of at least three biological replicates. (** *p* < 0.01 by two-tailed Student’s *t*-test).

**Figure 3 ijms-23-08382-f003:**
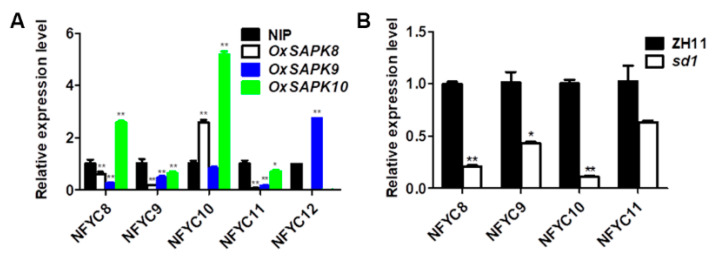
(**A**) The expression level of *NFYC8* to *NFYC12* in the seeds of wild-type-NIP, *OxSAPK8*, *OxSAPK9,* and *OxSAPK10* mutants that germinated for 6 h. (**B**) The expression level of *NFYC8* to *NFYC11* in the seeds of wild-type-ZH11 and *sd1* mutant that germinated for 6 h (* *p* < 0.05, ** *p* < 0.01 by two-tailed Student’s *t*-test).

**Figure 4 ijms-23-08382-f004:**
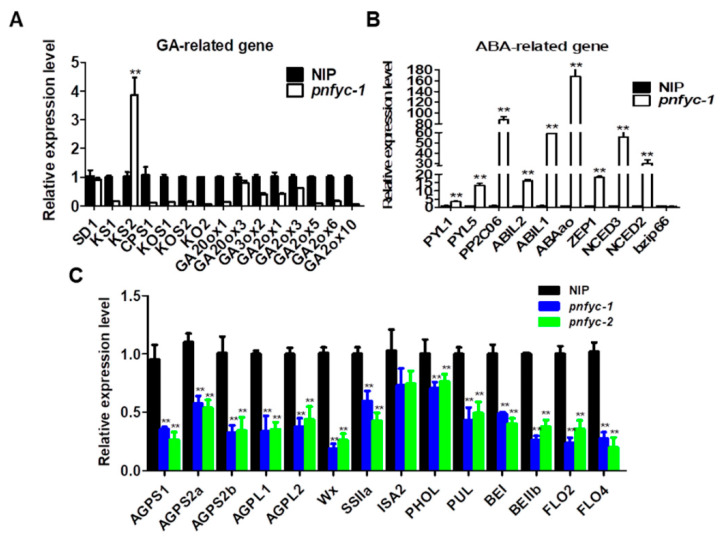
(**A**) The expression level of GA synthesis and metabolism-related genes in the seeds of wild-type-NIP and *pnfyc* mutants that germinated for 6 h. (**B**) The expression level of ABA biosynthesis and negative signal pathway-related genes in the seeds of wild-type-NIP and *pnfyc* mutants that germinated for 6 h. (**C**) The expression level of starch synthesis-related genes in wild-type-NIP and *pnfyc* mutant seeds 7 days after pollination. Data are shown as means ± SD of at least three biological replicates. (** *p* < 0.01 by two-tailed Student’s *t*-test).

**Figure 5 ijms-23-08382-f005:**
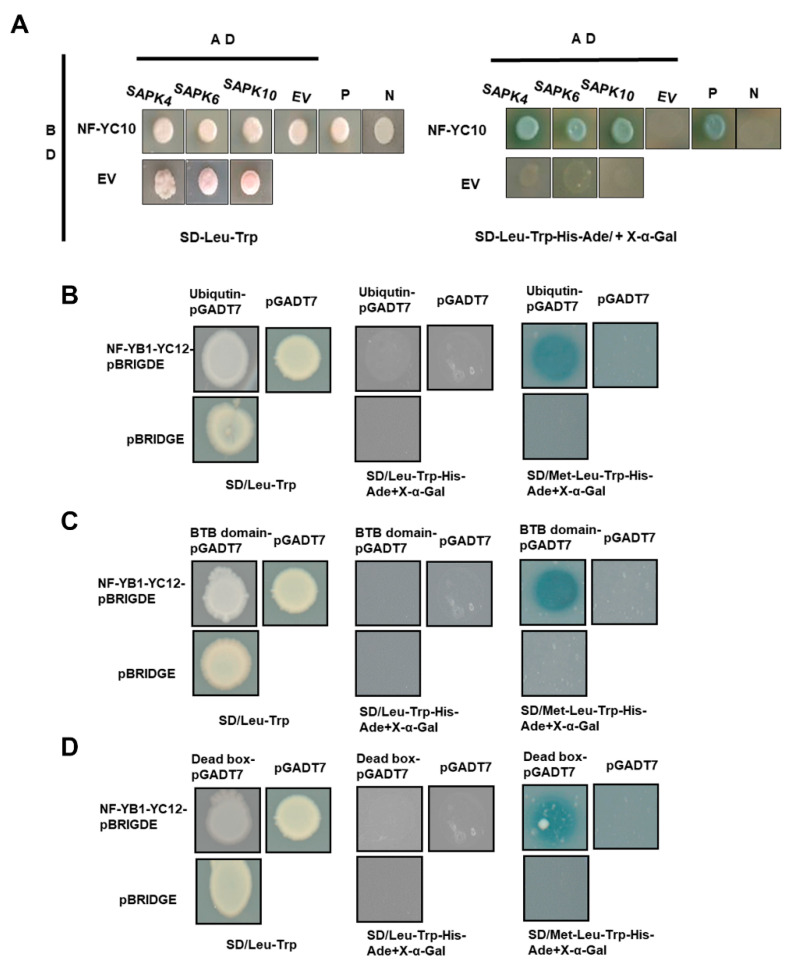
(**A**) Y2H assay of interaction between NF-YC10 and SAPKs. BD: pDEST32; AD: pDEST22; EV: empty vector, pDEST32 or pDEST22; P: positive control, pGBKT7-53/pGADT7-T; N: positive control, pGBKT7-Lam/pGADT7-T. (**B**) Y3H analysis of NF-YB1, NF-YC12, and Ubiquitin protein domain. (**C**) Y3H analysis of NF-YB1, NF-YC12, and BTB/POZ protein domain. (**D**) Y3H analysis of NF-YB1, NF-YC12, and Dead-box protein domain.

## Data Availability

Data is contained within the article and within Appendix A.

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
