# Peer review of "Five Rice Seed-Specific NF-YC Genes Redundantly Regulate Grain Quality and Seed Germination via Interfering Gibberellin Pathway"

_ijms, 2022, doi:10.3390/ijms23158382_

Round 1

Reviewer 1 Report

The authors aimed to determine the role of the transcription factors NF-YC8, 9, 10, 11, and 12, which are specifically expressed in seeds, during seed development. The authors generated these quintuple mutants and analyzed their endosperm traits, germination-related traits, and gene expression. As a result, the authors claimed that NF-YC8, 9, 10, 11, and 12 affect seed quality and germination through the ABA and GA pathways. 

The data on NF-YCs on endosperm will provide new insights for future seed quality control. However, below I have several concerns that the authors may consider in the further process of improving the manuscript.

Major points

1.     Lines 148-158.

Even though this manuscript aims to evaluate the redundancy of NF-TCs, this part of the manuscript only shows results for single mutants. The authors should also present similar data in quintuple mutants.

2.     Lines 218-235.

The authors have shown in Fig. 3 that NF-YCs may affect the expression of SARKs. On the other hand, this paragraph abruptly verifies the protein interaction between NF-YCs and SARKs. The authors should add further explanations, including the intent of the experiment.

3.     The discussion in every paragraph is simply a report of a previous paper. Furthermore, there is no conclusion to the manuscript, and it is unclear what the authors are ultimately trying to argue.

4.     Why does the title only include GA? Should it also describe ABA.

Minor point

5.     Line 98. 

Can the authors show the data of plant height and flowering date?

Reviewer 2 Report

The idea of generating a pentuple mutant and comparing it with single mutant to shed new light into the gene function of NF-YCs is appreciable. To improve the overall quality of manuscript following concerns should be addressed:

1.     In the introduction, there are many sentences which seems incomplete for example Line:41-44, it is obvious in an important crop like rice to have enough of literature for numerous genes controlling seed development therefore it is better to cite and explain the research available for NF-YC.

2.     In the results or methods explain about the genic regions selected as target of CRISPR and basis for selecting those regions. I think if these genes are highly similar how you have separated the mutant sites from each other.

3.     Have you studied the starch granules in individual mutants also if yes it will be interesting to know the differences and compare them with pentuple mutants? If you have available data for them, please add that in main or supplementary information.

4.     In the discussion, it is important to have a conclusion statement which summarize overall idea and importance of the study and provide future perspective.
